# Novel Vision Monitoring Method Based on Multi Light Points for Space-Time Analysis of Overhead Contact Line Displacements

**DOI:** 10.3390/s22239281

**Published:** 2022-11-29

**Authors:** Andrzej Wilk, Len Gelman, Jacek Skibicki, Slawomir Judek, Krzysztof Karwowski, Aleksander Jakubowski, Paweł Kaczmarek

**Affiliations:** 1Faculty of Electrical and Control Engineering, Gdańsk University of Technology, Narutowicza 11/12, 80-233 Gdańsk, Poland; 2Department of Engineering and Technology, School of Computing and Engineering, University of Huddersfield, Queensgate, Huddersfield HD1 3DH, UK

**Keywords:** electric traction, overhead contact line vibration, vision measurements, image processing

## Abstract

The article presents an innovative vision monitoring method of overhead contact line (OCL) displacement, which utilizes a set of LED light points installed along it. A light point is an, LED fed from a battery. Displacements of the LED points, recorded by a camera, are interpreted as a change of OCL shape in time and space. The vision system comprises a camera, properly situated with respect to the OCL, which is capable of capturing a dozen light points in its field of view. The monitoring system can be scaled by increasing the number of LED points and video cameras; thus, this method can be used for monitoring the motion of other large-size objects (e.g., several hundred meters). The applied method has made it possible to obtain the following novel results: vibration damping in a contact wire is nonlinear by nature and its intensity depends on the wire vibration amplitude; the natural frequency of contact wire vibration varies, and it is a function of vibration amplitude; the natural frequency of contact wire vibration also depends on the wire temperature. The proposed method can be used to monitor the uplift of contact and messenger wires in laboratory conditions, or for experimental OCL testing, as well as for verifying simulation models of OCL.

## 1. Introduction

An overhead contact line (OCL) is still the most effective way to provide power to electric railway vehicles. The speed of the moving vehicles is increasing, which results in the power demand escalation and subsequent problems with meeting the requirements ensuring correct cooperation of vehicle pantographs with OCL [1,2,3].

OCL and the electric vehicle pantograph compose an electromechanical system, which is crucial for the course of changes of the contact force between the OCL and the pantograph head [4,5]. Ensuring the correct interaction of these two elements requires a proper design of the OCL structure, taking into consideration mechanical, electrical, and environmental impacts. In particular, technical requirements, which are defined in Technical Specifications for Interoperability (TSI) [6] should be met. One of these requirements refers to the displacement range of the OCL contact wires due to the interaction of the pantograph during the motion of a vehicle. The propagation of longitudinal and lateral vibrations occurs due to mechanical excitations and wave reflections at OCL fixing nodes. The displacement of the contact wire from its steady state takes place in a certain spatial area, which changes with time. Hence, measuring this displacement is a relatively difficult task, as the scale of the contact wire uplift is up to several tens of millimeters over the length of about several tens of meters. The wire vibration range in the time domain is also relatively wide and amounts from several tens to several hundreds of seconds. Moreover, the contact wire in operating conditions has a relatively high electrical potential, for instance, 3 kV DC or 25 kV AC, which requires galvanic separation of measuring sensors. Various diagnostic and monitoring systems are developed to keep OCL in proper technical condition during operation [7,8,9,10,11]. The search for reliable, safe, inexpensive, and precise measuring and monitoring methods is still ongoing [12,13,14,15,16,17,18].

Figure 1 shows a sketch of electric railway traction, with the highlighted OCL—pantograph system.

The dynamics of the pantograph’s interaction with OCL is being intensively studied, especially for high-speed railways. Various theoretical analyses of the problem are presented in the literature—starting from relatively simple, based on the model of current collector with one degree of freedom and periodically changing flexibility of OCL, up to complex models making use of modal analysis, finite difference method (FDM) or finite element method (FEM), hybrid methods, etc. [3,4,13].

In accordance with TSI recommendations and relevant normative measures, the pantograph—OCL interaction should be evaluated via simulation concerning such parameters as: contact wire uplift, mean value and standard deviation of contact force, and others.

At present, the models of current collectors are already worked out in detail [19,20,21,22,23], but there is still a need for simulation models of OCL focused on monitoring current collectors in real operating conditions [21,22,24] and making use, for instance, of digital twin methodology [25,26,27].

The accuracy of simulation results is highly dependent on the proper parametrization of a model. One way to obtain valid parameters is to conduct experimental tests, under laboratory or operational conditions. In the case of the overhead contact line, the length of the catenary suspension span is typically within several tens of meters.

One of the possible solutions allowing for the effective measurement of parameters for dynamic systems of large dimensions is based on vision measurement technology. Such an approach was applied to parameter identification of rotor blades of wind turbines, cable—stayed bridges, structures, and, finally, railway overhead contact line. In ref. [28], a multi-camera photogrammetric system was presented for simultaneous measurement of deformation and displacement of a wind turbine blade.

Other solutions propose the measurement of suspended structure vibrations, that rely on recording the displacement of a single construction element, with the bridge and the stadium canopy used as examples [29,30]. However, measurement in subsequent points of the object requires relocation of the camera setup and, consequently, a time-consuming calibration is necessary. Vision measurements were also used for the identification of building structural models parameters [31,32,33,34].

Researchers also used reflectors fixed at distinctive points of the object [33]. The reflected light might be in the visible or infrared range. At the same time, the displacement of a single reflector was recorded and analyzed. In [35] was presented a method for determining the parameters of an OCL by measuring the wire displacement with a vision system. The concept was based on a simultaneous recording of the displacement of two points of connection of the dropper with the messenger wire and contact line (Figure 1). For this purpose, two cameras synchronized with an external signal were used. In order to analyze the entire suspension span, the system was moved every few meters in a way that allowed to maintain the mutual geometric relationships between the camera and the observed points. The measurements were carried out on a railroad line section in normal operating conditions. Due to the lack of possibility of simultaneous measurement of displacements at multiple points of the span, measurements were repeated many times in order to record the results of the different trains passing. Such conditions, however, do not ensure repeatability of excitations for the analysis of displacements of the catenary along the whole suspension span. For this reason, a statistical analysis of obtained results was proposed.

Another stereo vision method proposed for measuring only the static geometry of OCL is described in [8,9,14,15,36].

In order to eliminate the shortcomings of the described methods, an innovative vision method is proposed in this article to monitor dynamic displacements of the contact wire using light points, installed along it. A light point is an individually fed light emitting diode (LED) of relatively small dimensions, as compared to the range of the observed displacements. Further in the article, this method is referred to as 4D, which means three spatial coordinates *x*, *y*, *z* of contact wire position, and time *t*. The *x*—coordinate describes the distribution of LED points along the overhead contact line.

The significant advantages of the proposed method compared to other methods are the determination of the shape of the contact wire at steady and transient states and over a span length with the use of a single vision camera. The system is scalable by adding light points and vision cameras. Obtained results are necessary for the analysis of the space-time distribution of the contact wire vibration mode. This is useful for the parametrization and validation of the mathematical models of OCL [37,38,39]. The proposed method can apply both in the laboratory and operational measurements on a railway line.

The overhead contact line is a complex electromechanical system that usually consists of one of two contact wires connected with the messenger wire via droppers. The entire system is mounted on the support structures using insulated brackets [1,2]. The OCL is subject to spatial adjustment with an accuracy of 1 cm. Selected courses of changes of OCL suspension height and stagger are shown in Figure 2. These quantities were measured in the static state of OCL operation using the vision method and a camera with a linear illuminator mounted on a measuring vehicle [7]. The applied vision method is a variant of the method proposed in Section 2.

Poor technical condition of the contact wire may cause higher oscillations and locally more intensive wear, which in extreme cases would lead to the failure of the cooperating pantograph. The task of the diagnostic and monitoring systems is to detect degradation of the technical condition of OCL to prevent damage and avoid extra costs, also those resulting from disturbances in railway traffic. The applied measuring methods should be based on the analysis of parameters that can be relatively easily measured [36,39].

With respect to pantographs, their technical condition is controlled, for instance, at selected OCL suspension points by indirect measurement of contact wire uplift during vehicle movement under a certain checkpoint [5]. Typical values of contact wire uplifts, measured using a laser distance meter at a selected OCL suspension point, are shown in Figure 3 for different situations of motion. Different waveforms of contact wire uplift, accompanied by OCL vibrations excited by the interaction of the pantograph, can be observed. The shape of the waveform depends on pantograph contact force, number and distance between pantographs, and vehicle speed, among other factors.

Another important issue related to OCL operation includes tension and wear of contact wires. The effect of the tension force is well described in OCL models and operating conditions. However, the friction forces acting between the contact wire and the pantograph contact strips cause a gradual reduction in the wire cross-section. The wear of the wire leads to a decrease in its mechanical strength and an increase in electrical resistance. Figure 4 shows vibration waveforms recorded in the laboratory using a laser distance meter for a brand-new wire and the wire with uniform wear of about 15% (measured in the cross-section area) to which the same tension force was applied (experimental results obtained by the authors of this article [7]). The analysis of the recorded vibration waveforms indicates that a high degree of contact wire wear increases the amplitude of wire vibration along the lateral axis, which becomes comparable to that of vertical vibration. It is noteworthy that the vibration parameters are also affected by the twisting of the worn wire and contact wire stagger at OCL suspension points.

Monitoring systems making use of a digital twin concept are introduced on main railway lines. However, currently used systems of this type are very complicated and expensive [25,26,27]. Therefore, there is a need to develop simpler and much cheaper monitoring systems viable for less important lines. The main component of such a system is a digital twin, i.e., the model which adapts to changes of operating conditions based on the online collected data and can predict in real time the technical condition of its physical equivalent. For this purpose, physical factors should be considered in the OCL simulation model. To complete this task, a monitoring method has been developed and experimental tests performed, as described in further Sections of this article.

In our novel method, compared with the well-known methods, the following novel steps/operations are proposed:determination of the shape of the contact wire at transient and steady states over a span length (several dozen meters) with the use of a single vision camera,scalability over hundreds of meters by adding another camera and light points,possibility of application in various ambient light conditions due to active LEDs,robust and fast algorithm for detecting the position of a LED point.

As a result of the implementation of the proposed method, the following novelties were obtained, which have not been reported so far:novel experimental results of OCL vibrations for different temperatures,novel experimental research results of the impact of the vibration amplitude and wire temperature on the damping properties of the OCL and its natural frequency based on the proposed method.

This paper aims to achieve the following objectives:monitoring of dynamic behavior of OCL using a single camera and set of LED points,improvement of measurement scope: the method allows for scalability of the monitoring system and can be useful for longer OCL fragments,improvement of measurement accuracy: obtained results can be used in the development of digital twins for OCL.

The article shows the original results of contact wire vibration monitoring in a transient state resulting from the rapid removal of the imposed force in the direction perpendicular to the wire longitudinal axis. Section 2 describes the concept of 4D monitoring of contact wire displacements and the test stand with necessary instrumentation. Section 3 presents the methodology of experimental tests of wire vibration dynamics. Section 4 describes the methodology used for processing and analyzing the data obtained from the test stand and presents the original results of contact wire displacement in time and space. Section 5 discusses the obtained results and presents the conclusions resulting from the application of the innovative vision system making use of light points.

## 2. Vision Monitoring Method

The proposed vision method to monitor a selected light point *P*(*z*, *t*) moving in time and space is based on a principle of optical mapping, schematically shown in Figure 5 [7,35].

For the situation shown in Figure 5, the displacement of the light point *P*(*z*) at position z, can be monitored based on its image *P*′(*z*′) on the camera matrix plane and characteristic spatial dimensions *k* and *F* of the system, according to the following formula:(1)$$z=z^{\prime }\cdot \left(\frac{k}{F}-1\right),$$
where *k* is the distance between the central point on the object plane and the image plane, and *F* is the distance between the optical center of the lens and the image plane.

The relationship for observations along the perpendicular *y*-axis is the same. The digital camera matrix is two-dimensional (2D) which makes it possible to obtain information about the position of light point *P*(*y*, *z*) and, at the relevant recording speed of the camera, the values *P*(*y*, *z*, *t*) describing object displacement in time in 2D space.

The spatial configuration, shown in Figure 5, cannot be directly used for monitoring the displacement of the OCL contact wire and should therefore be modified. In particular, parallel placement of the observation plane (object plane with LED point) and the matrix surface (image plane) should be changed in the way shown in Figure 6 [7].

To take into account the presence of deformation characteristic for optical foreshortening, the mathematical relations used to calculate the LED point displacement (and, consequently, the displacement of the contact wire) take the following form:(2)$$y=\frac{\left(k-F\right)\cdot y^{\prime }\cdot \cos\beta }{\cos\alpha \cdot \left(F\cdot \cos\beta -z^{\prime }\cdot \sin\beta \right)-y^{\prime }\cdot \sin\alpha },$$
(3)$$z=\frac{\left(k-F\right)\left(z^{\prime }+y^{\prime }\cdot \sin\beta \cdot \tan\alpha \right)}{F\cdot \cos\beta -z^{\prime }\cdot \sin\beta -y^{\prime }\cdot \tan\alpha },$$
where *y*′, *z*′ are the coordinates of the image of measuring point in the image plane, *α* is the deflection angle between the plane with the lens optical axis and the plane with the contact wire, and *β* is the camera inclination angle from the horizontal, *k* is the distance between the central point on the object plane and the image plane, and *F* is the distance between the optical center of the lens and the image plane.

The vision system utilizing the principle, shown in Figure 6, allows to monitor LED point positions on the contact wire in the 2D plane. Then, the sequential recording makes it possible to observe changes of light point positions in time. To increase a measuring range and to allow monitoring position changes in the third spatial dimension, i.e., for successive LED points situated along the contact wire, the system shown in Figure 7 should be applied.

To conduct the monitoring with the proposed method, the camera should be equipped with a lens with marginal optical defects. The view angle of this lens should allow capturing all LED light points along the contact wire. The image of each light point is mapped to a corresponding point on the camera matrix. The measurement results for each point are obtained from relations (2) and (3), bearing in mind that the values of the distance *k* and the angles *α* and *β* are different for each light point. The lens aperture should be selected in such a way that the depth of field will cover all LED points, and the focus should be set to the hyperfocal distance.

In this way, the 2D image camera can be used for monitoring 3D positions of light points. A sequence of these LED points recorded by a digital camera represents a discretized shape of the contact line in 3D space. The distance between the LED points ranging from a fraction of to a few meters is sufficient for unambiguous determination of the vibration mode of a typical OCL system. It is assumed that one video camera can simultaneously cover a space with more than ten light points. Selecting a suitable location for the camera makes it possible to determine the coordinates of the light points in time and space with sufficient accuracy and resolution [7]. Consequently, by recording time-dependent changes of position parameters we obtain an innovative 4D vision monitoring system. The application of this system and making use of its basic advantage, which is synchronic position monitoring of a number of points along the contact wire, allows to perform a relatively simple study of, for instance, propagation of mechanical waves, which would be a very complicated task when using traditional measurement methods.

## 3. Experimental Test Stand

Based on the above-presented assumptions for the novel 4D vision method, an experimental test stand was built to monitor contact wire vibrations. A contact wire of 26 m in length was tensioned between support structures. Nine light points in the form of LEDs emitting light were installed along the contact wire. To preserve a small mass of individual light points, each LED was fed individually from a battery. Thus, the mass of LED points is negligibly small compared to the mass of the monitored object. The scheme of the test stand is shown in Figure 8.

For the geometrical configuration of the stand from Figure 8, the measurement uncertainty was estimated. It was assumed that all dimensions are determined with a standard uncertainty not worse than 1 mm. Under this assumption, the measurement uncertainty does not exceed 4 mm. The measurement uncertainty was estimated according to the procedure described in [40] and on the basis of the principles presented in [41]. Further analysis shows that when using a lens with a long focal length is possible to obtain a measurement uncertainty of 1 mm.

The view of the test stand is shown in Figure 9. The light points are visible in the form of shining LEDs. The laser distance meter allowed to verify the results of the LED point position monitoring was applied. In order to process recorded images, authors developed original software using the LabVIEW environment. Typical analysis modules were used for image processing, i.e., thresholding, partial filtration and dilation [42,43,44]. Digital filtering of the raw measurement results was performed by using the Whittaker algorithm [45]. The application converts the graphical 2D data into numerical 3D light points coordinates that can be filtered, plotted, or exported for further analysis. The code has been optimized to ensure good computation performance on a standard PC platform. Detailed information related to the applied image processing algorithm can be found in [42,43,44].

The contact wire was tensioned via a typical construction, used on railway lines (Figure 10). The tension force applied to the wire in the resting position amounted to 10 kN. The force was measured with a strain gauge, and the measured results were stored synchronically with the wire displacements.

The contact wire displacement was recorded using the Basler acA2040—180kc camera with a resolution of 2046 × 2046 pixels (4 Mpix) and sensor dimensions of 11.26 × 11.26 mm. Successive picture frames were recorded with a speed of up to 180 fps. The camera was equipped with a Lydith 3.5/30 lens, with a focal length of 30 mm and light-gathering power of 1:3.5. The used lens was characterized by negligibly small optical defects in the camera image field. The camera was connected to the computer via the Camera Link interface.

The camera was positioned in the way shown in Figure 7 to make all LED points visible in the picture frame. A selected picture frame recorded in the computer is shown in Figure 11.

At the midpoint of the contact wire length, a force of 100 N was imposed using a weight suspended by a connecting rod. Vibrations of the wire were excited by removal of the weight. The subsequent data acquisitions were performed for the following steady—state temperatures of the contact wire: 27.7 °C, 29.7 °C, 41.4 °C, 62.8 °C, and 73 °C. The temperature was measured using an electronic thermometer with a thermocouple sensor. The source of the heat released in the contact wire was the power loss caused by wire electrical resistance. The contact wire temperature was controlled by changing the current flow through the wire.

## 4. Selected Results and Their Analysis

This section presents selected monitoring results of vertical displacement of a brand-new OCL contact wire with cross-section area of 100 mm^2^. Vibration waveforms related to light points *P*_1_(*z*, *t*), …, *P*_9_(*z*, *t*) are shown in Figure 12. light points were installed at a constant interval of 2 m, as shown in Figure 8. The horizontal axis *x* represents light points placement from *x*_1_ to *x*_9_. Time value *t* < 0 represents the steady-state position of the contact wire under the load of the imposed vertical force. Vertical axis *z* shows the values of the displacement for each LED point. At time *t* = 0 step removal of the force occurred, and the oscillations of the contact wire has begun. The two initial periods of oscillations are shown in Figure 12. Moreover, every 0.1 s line connecting the positions of the LED points were plotted, revealing the dynamically changing shape of the contact wire. The positions for *t* = 0 and *t* = 1.1 s have been emphasized. Horizontal oscillations in the *y*-axis are not shown, as their amplitude is negligibly small in comparison to oscillations observed in the *z*-axis.

It is worth emphasizing that the proposed method allows for simultaneous registration of all light points positions, which is an advantage over existing measurement methods.

Waveform of displacement for point *P*_5_(*z*, *t*) situated at the distance of 1 m from the excitation force application point (see Figure 8) is shown in Figure 13. The data acquisition was performed for wire temperature *T* = 27.7 °C. The qualitative analysis of changes of displacement *z*(*t*) for selected time intervals reveals the presence of nonlinear damping.

Detailed view of the waveform from Figure 13 is shown in Figure 14 at a relatively short time window.

Figure 15 shows displacements along *z*-axis as functions of light point position *x*_k_ (k = 1, 2, …, 8). It was assumed that the *x*-component is equal to zero at the excitation force application point situated between points 4 and 5. The figure also presents the instantaneous shapes of the contact wire for selected times. The time *t* = 0 s was assumed the initial condition for the transient state caused by the rapid removal of the imposed force. The time stamps were selected in such a way as to correspond to the occurrence of the first and sixth maximum and minimum. Figure 15a shows the results for steady-state temperature of the contact wire equal to 27.7 °C, while Figure 15b for temperature of 73.0 °C.

Figure 16 shows the relationship between *z*-component of wire displacement and natural frequency *f* determined at the light point *P*_5_ for different values of contact wire temperature. Changes in natural frequency as a function of contact wire temperature changes are very subtle, especially for higher temperatures. The intersection of characteristics for temperature 41.4 and 62.8 °C is unexpected. The probable cause of this phenomenon is measurement uncertainties, but further research is required to determine this.

The analysis of phenomena based on a mathematical model requires the use of an appropriate system of differential equations. In this work, the authors propose a qualitative analysis based on a relatively simple model of a one-dimensional damped harmonic oscillator [46]. Qualitative analysis means that the obtained results based on the simulation of the model do not have to be consistent with the measurement results. The authors assumed, however, that a one-dimensional nonlinear harmonic oscillator will show the following relationships that are observed in a real system:influence of the non-linearity of the damping coefficient on the waveform of the displacement,influence of the non-linearity of the damping coefficient on the angular frequency of the displacement,influence of the spring coefficient on the angular frequency of the displacement—an indirect way of assessing the influence of temperature on the frequency of vibrations.


The equation for a one-dimensional free oscillator takes the form:
(4)$$\ddot{z}\left(t\right)+\gamma \dot{z}\left(t\right)+\omega_{0}^{2}z\left(t\right)=0,$$
where: *γ* is damping coefficient, *ω*_0_ is natural angular frequency, $\omega_{0}^{2}$ represents force per unit displacement per unit mass.

The solution of this equation at constant parameters is given by:(5)$$z_{1}\left(t\right)=e^{-0.5\gamma t}\left\{z_{1}\left(0\right)cos(\omega_{1}t)+\left[{\dot{z}}_{1}\left(0\right)+\frac{1}{2}\gamma z_{1}\left(0\right)\right]\frac{\sin\left(\omega_{1}t\right)}{\omega_{1}}\right\},$$
where: *z*(0), *ż*(0) are the initial conditions of displacement and velocity, respectively.

The angular frequency is given by
(6)$$\omega^{2}=\omega_{0}^{2}-\frac{1}{4}\gamma^{2},$$

From Figure 13 it can be concluded that the damping effect is relatively small for times greater than about 80 s—the amplitudes of successive oscillations vary relatively little with respect to each other. Hence, the value of natural angular frequency $\omega_{0}$ in Equation (4) was assumed to be the same as the value of angular frequency $\omega_{1}$ = 12.389 rad/s at time *t* = 80 s calculated from experimental data.

In order to show the influence of the non-linearity of the damping coefficient on the waveform of oscillator vibrations, two cases were analyzed. In the first case, the coefficient *γ* was assumed as a constant value, while in the second one it was dependent on the value of the vibration velocity *ż*(*t*).

In the first case, the constant damping parameter *γ* was determined from the iterative procedure in such a way that the displacement amplitude of oscillation obtained from the simulation and measurement was the same for the time *t* = 80 s. At *γ* = 0.077 Nsm^−1^ kg^−1^ the magnitude of displacement obtained from the simulation is 2.5 mm at *t* = 80 s and is the same as the magnitude obtained from the experiment.

In the second case, the damping parameter *γ* = f(*ż*(*t*)) was assumed as a function of vibration velocity. Using the iterative procedure this function has been determined in such a way that the displacement amplitude of oscillation obtained from the simulation and measurement was the same (2.5 mm) for the time *t* = 80 s. When looking for the dependence *γ* = f(*ż*(*t*)), only linear dependences of the type *γ* = *a·ż*(*t*) were analyzed. A good agreement of the envelope between the simulation and measurement waveforms was obtained for the value of the coefficient *a* = 0.91 Ns^2^m^−2^kg^−1^.

Figure 17, Figure 18, Figure 19 and Figure 20 show the simulation results of these two cases-linear (Figure 17 and Figure 18) and nonlinear (Figure 19 and Figure 20). The differential Equation (4) was solved (linear and nonlinear case) by the Runge-Kutta numerical method implemented in the Matlab environment.

Detailed view of the waveform from Figure 17 is shown in Figure 18 at a relatively short time window.

Detailed view of the waveform from Figure 19 is shown in Figure 20 at a relatively short time window

It should be clearly emphasized that the one-dimensional oscillator model does not correspond to the complex model of the analyzed contact wire. However, a good agreement of the envelope between the simulation (one-dimensional oscillator) and measurement (point *P*_5_) waveforms indicates non-linear damping properties occurring in the real system. When both values of velocity and displacement are relatively high the damping property of the contact wire is also relatively high.

From Equation (6) one can conclude that the angular frequency is increased when the value of the damping coefficient is decreased. This formula explains the increase in angular velocity with a relatively low value of wire displacement associated with a low vibration velocity.

## 5. Conclusions and Future Research Perspectives

The paper proposes an innovative method to perform space-time monitoring of lateral displacement of a contact wire with the use of miniature light points and a vision camera, along with relevant novel software. The proposed method makes it possible to:Monitor displacements of discrete points situated along the contact wire in the time domain, within the range depending on the number of LED points and vision cameras—the system is scalable;Monitor wire displacements in experimental conditions of OCL operation. Like a dropper, the light point is fixed to the upper part of the contact wire and does not disturb the OCL—pantograph cooperation. The LED point can be installed at an arbitrary electrical potential of the OCL.

Based on the analyses performed both in time and space domains, the following original results were obtained with respect to transient states of contact wire vibrations:The vibration-damping phenomenon is nonlinear by nature. For higher vibration amplitudes, the damping coefficient per unit mass is greater than for lower amplitudes. The natural vibration frequency for higher amplitudes is slightly lower than that for lower amplitudes;The natural vibration frequency of a wire depends on its temperature. With increasing temperature, the natural vibration frequency clearly decreases, but only up to a given temperature. In the examined case, the recorded relation between temperature and natural vibration frequency became ambiguous. This aspect of contact wire behavior requires further studies;The proposed monitoring method allows to determine the contact wire shape for selected time stamps. The vibrating wire can take different shapes (modes), which indicates that the system is not symmetrical. For a given mode, higher spatial harmonics can be found, which can be explained by different boundary conditions at particular fixing nodes. One node had a degree of freedom in the wire axis direction, which resulted from the tension method applied to the wire.

The innovative method to monitor OCL contact wire displacement in time and space domains allows to obtain results within a relatively wide range of wire length and times in which the transient states take place. The method can be applied with relatively small measurement uncertainty–of the order of several millimeters. Using this method, original results on contact wire vibration have been achieved, and the nonlinear nature of relations between natural vibration frequency, vibration amplitude, and operating temperature of the wire has been identified, which testifies to the usefulness of the proposed method in studying dynamic OCL-pantograph interactions.

## Figures and Tables

**Figure 1 sensors-22-09281-f001:**
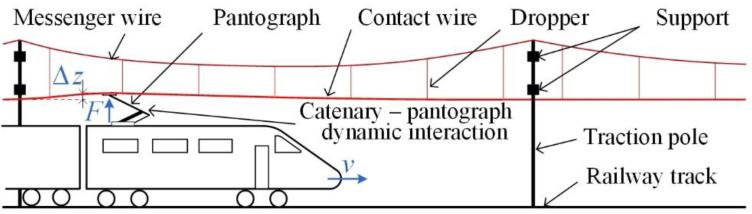
The OCL—pantograph system with marked: pantograph’s action with force *F*(*t*), contact wire uplift Δ*z*(*t*), and vehicle speed *v*(*t*).

**Figure 2 sensors-22-09281-f002:**
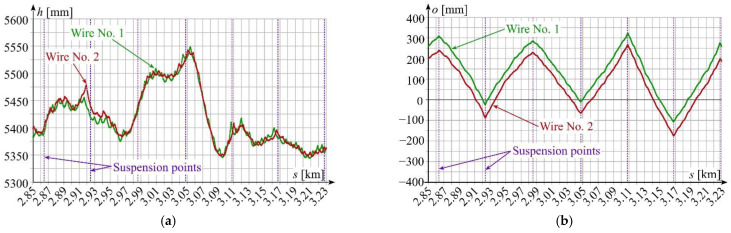
Overhead contact line DC 3 kV: (**a**) height *h*(*s*) of OCL suspension with two contact wires; (**b**) stagger *o*(*s*) of contact wires; source: own elaboration by the authors of this article.

**Figure 3 sensors-22-09281-f003:**
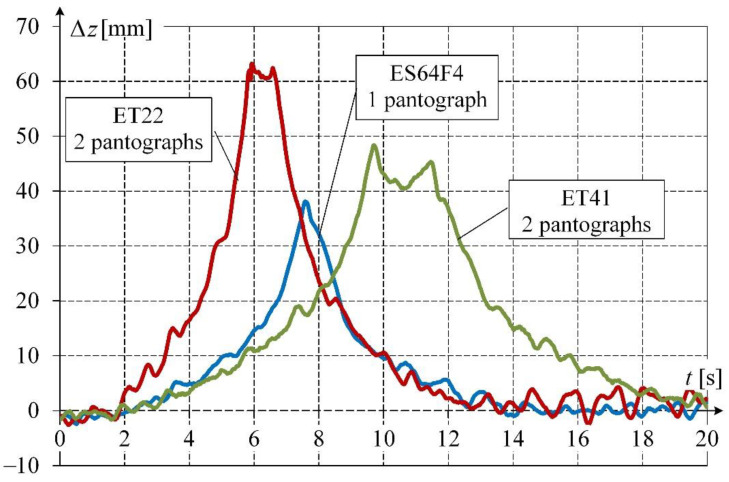
Selected waveforms of contact wire uplift Δ*z*(*t*) measured at OCL suspension point for different numbers of lifted pantographs of single-unit (ES64F4) and articulated locomotives (ET22, ET41); source: own elaboration by the authors of this article.

**Figure 4 sensors-22-09281-f004:**
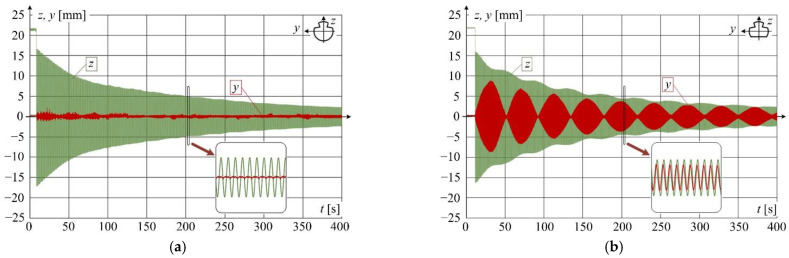
Vibrations of contact wire along *z*- and *y*-axes at 10 kN tension (*z*—green colour; *y*—red colour), for step change of vertical force of 150 N at ¼ of the 12-m contact wire span and vibration recording at ½ of wire span: (**a**) brand-new wire; (**b**) wire with a 15% wear rate.

**Figure 5 sensors-22-09281-f005:**
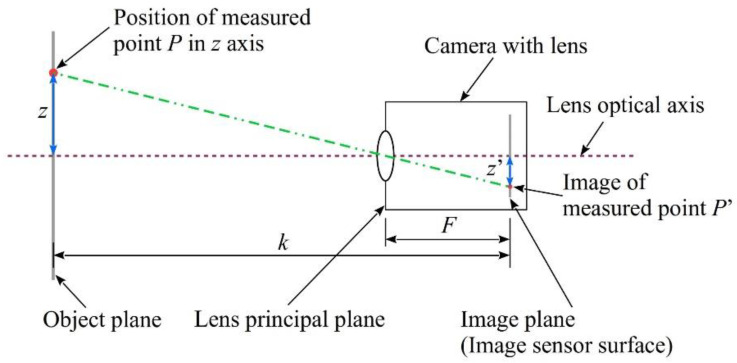
Principle of optical mapping—a basis for vision monitoring method.

**Figure 6 sensors-22-09281-f006:**
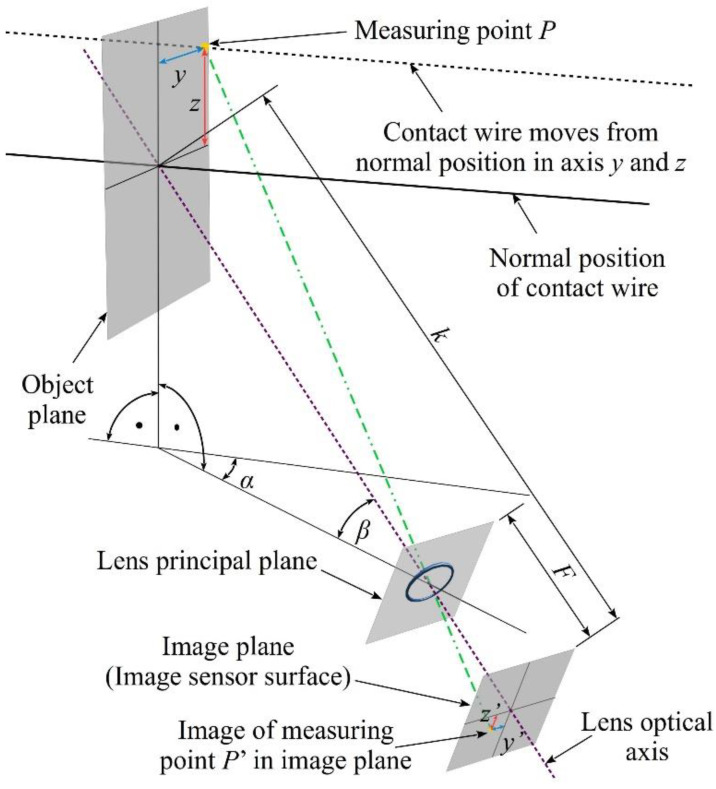
Spatial configuration of the vision method to monitor 2D displacement of *P*(*y*, *z*) in application to OCL.

**Figure 7 sensors-22-09281-f007:**
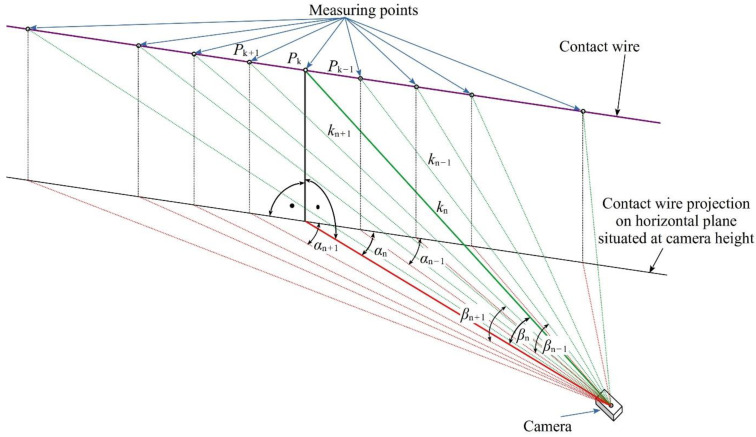
The principle of position monitoring of light points *P*_k_ on the contact wire in 3D space using 2D imaging camera.

**Figure 8 sensors-22-09281-f008:**
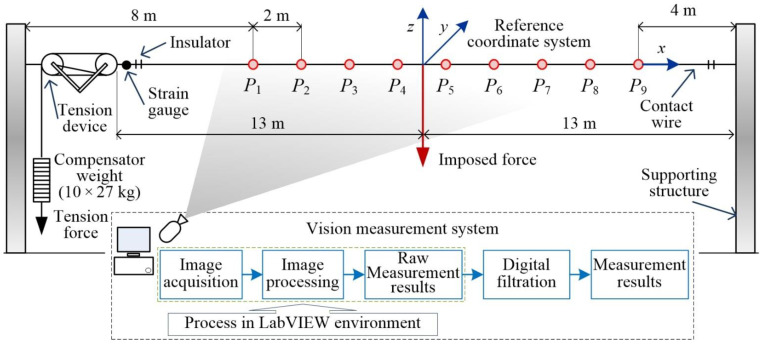
Scheme of the test stand, where *P*_1_, … *P*_9_—light points.

**Figure 9 sensors-22-09281-f009:**
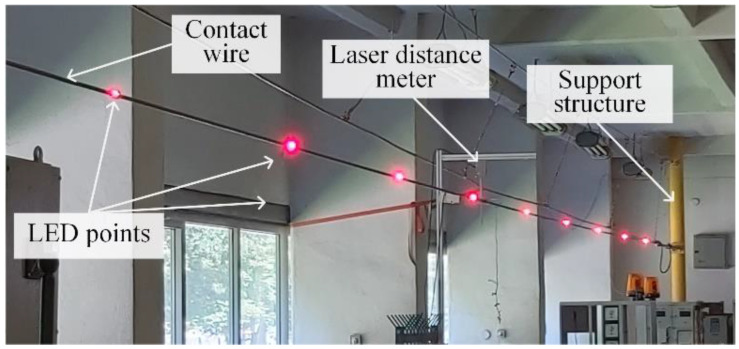
Contact wire with mounted LED points.

**Figure 10 sensors-22-09281-f010:**
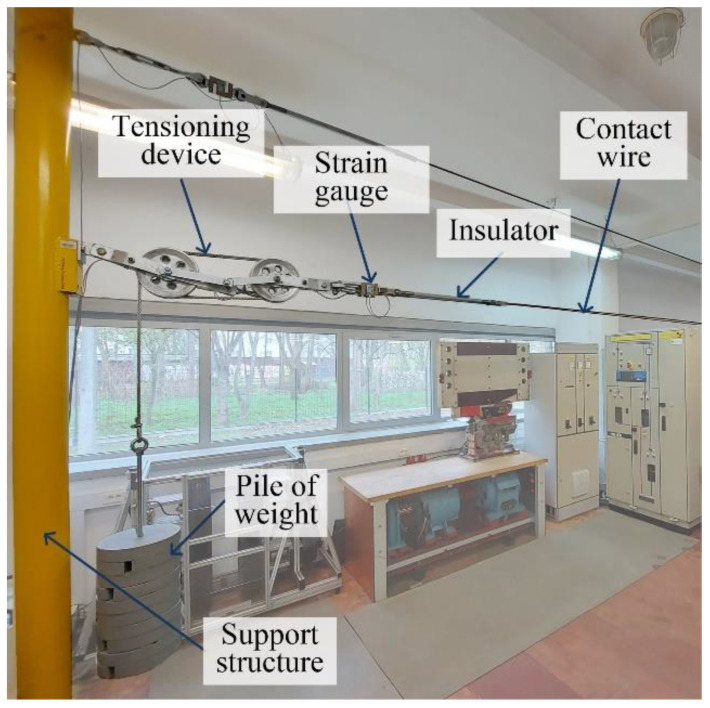
Tensioning construction-strain gauge for force measurement seen in the center.

**Figure 11 sensors-22-09281-f011:**
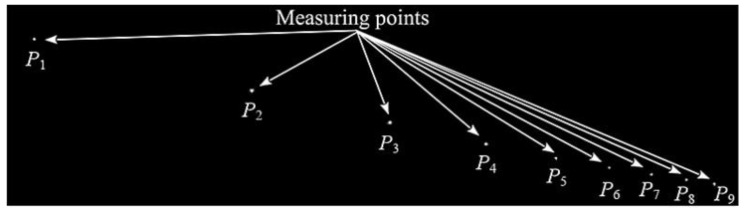
Selected image from the camera.

**Figure 12 sensors-22-09281-f012:**
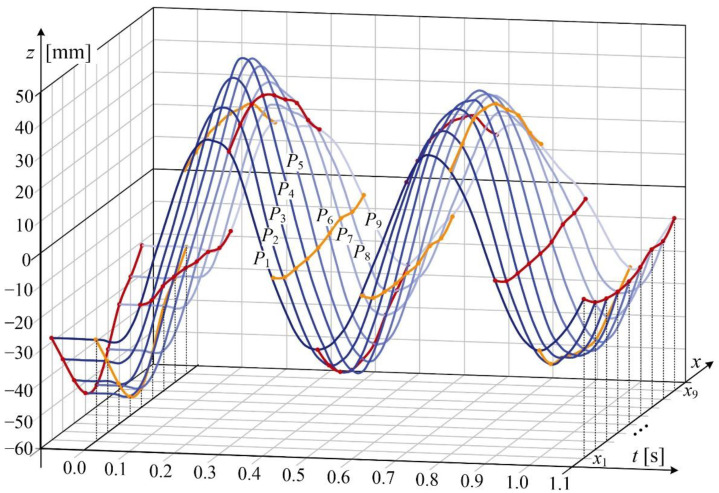
Displacement waveform of LED points. Red and orange lines visible position of the contact wire every 0.1 s.

**Figure 13 sensors-22-09281-f013:**
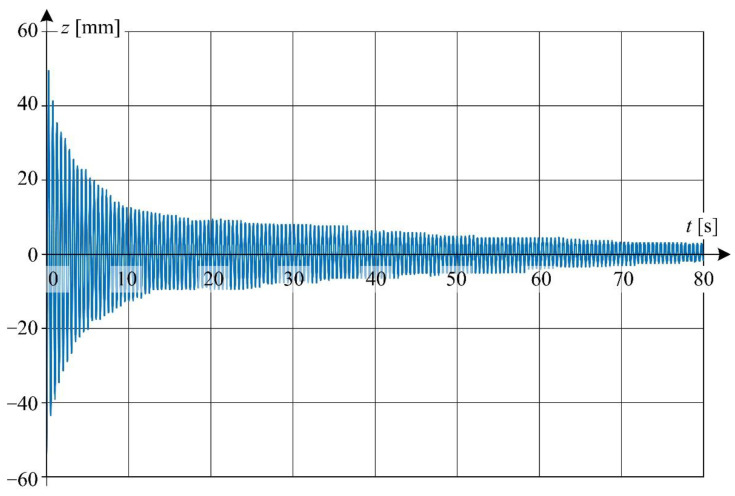
Displacement waveform of LED point *P*_5_(*z*, *t*) for wire temperature 27.7 °C.

**Figure 14 sensors-22-09281-f014:**
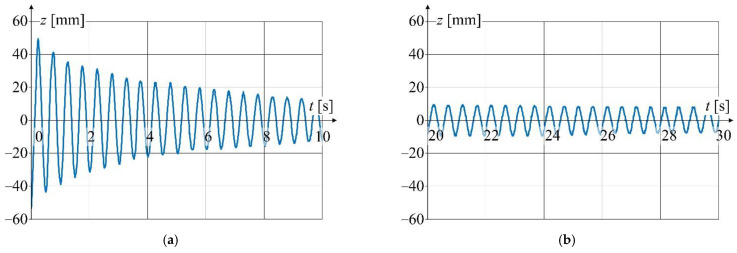
Detailed graphs of displacement waveform (Figure 13): (**a**) for the first 10 s of oscillation; (**b**) for 20 to 30 s of oscillation time.

**Figure 15 sensors-22-09281-f015:**
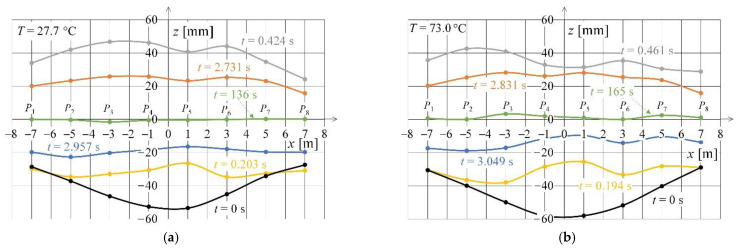
Spatial distribution of light points for selected recording times and contact wire temperature equal to: (**a**) 27.2 °C; (**b**) 73.0 °C.

**Figure 16 sensors-22-09281-f016:**
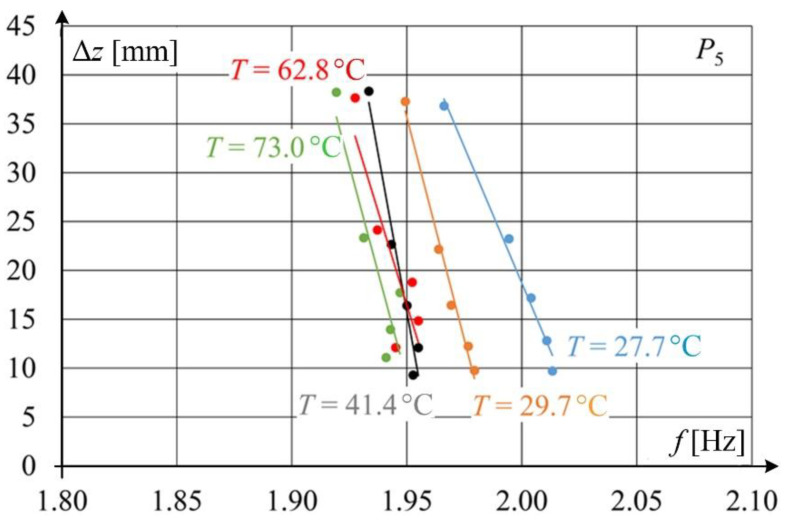
Relationship between natural frequency and vibration amplitude of the contact wire at different wire temperatures.

**Figure 17 sensors-22-09281-f017:**
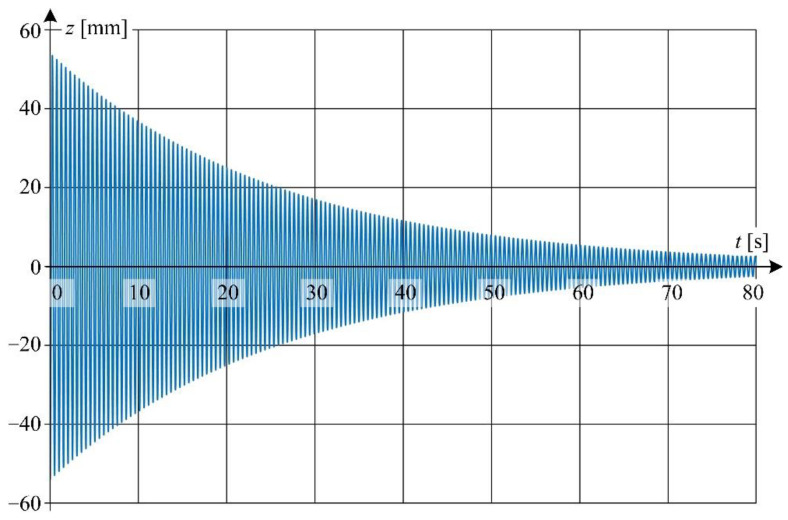
Displacement waveforms of simulation results at linear damping coefficient.

**Figure 18 sensors-22-09281-f018:**
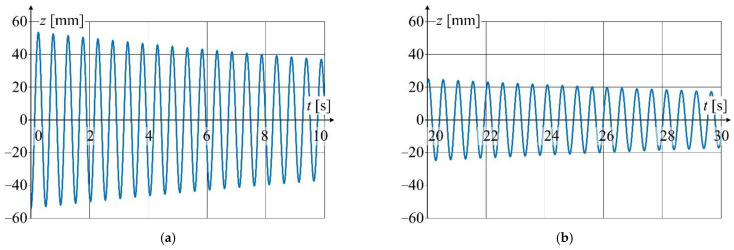
Detailed graphs of displacement waveform (Figure 17): (**a**) for the first 10 s of oscillation; (**b**) for 20 to 30 s of oscillation time.

**Figure 19 sensors-22-09281-f019:**
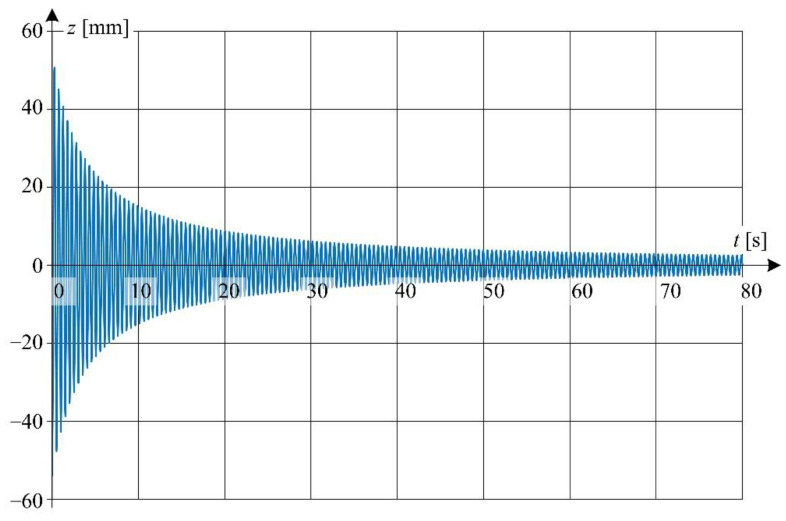
Displacement waveforms of simulation results at non-linear damping coefficient.

**Figure 20 sensors-22-09281-f020:**
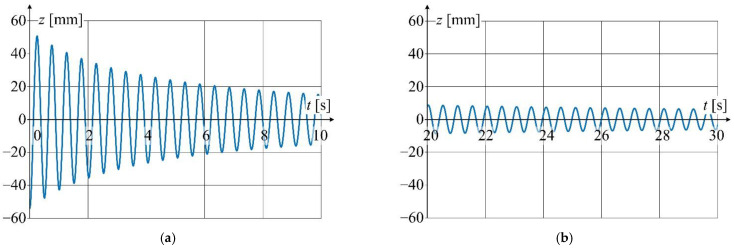
Detailed graphs of displacement waveform (Figure 19): (**a**) for the first 10 s of oscillation; (**b**) for 20 to 30 s of oscillation time.

## Data Availability

The obtained measurement data is confidential and therefore is not on a publicly accessible server.

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
