# Peer review of "Novel Vision Monitoring Method Based on Multi Light Points for Space-Time Analysis of Overhead Contact Line Displacements"

_sensors, 2022, doi:10.3390/s22239281_

Round 1

Reviewer 1 Report

The authors present an innovative vision monitoring method of overhead contact line (OCL) displacement. The topic of the manuscript is timely and perfectly falls within the scope of the journal. The methodology is presented in a sound fashion. The reviewer found some formal aspects that would require some revision before publication as indicated below:

1.     It is suggested that the authors provide quantitative descriptions to indicate the advantages of the proposed method in the abstract.

2.     The introduction is tediously long and confusing. It is recommended to simplify this section.

3.     Provide statistical indicators of displacement estimation results, such as error standard deviation.

4.     From Fig. 16, the fitting relationship curve moves to the left as the temperature increases. There is a cross between the red line and the black line. Explain the possible reasons.

5.     What means did the authors get the natural frequency? FFT or Welch?

6.     The conclusion is also suggested to be simplified.

Author Response

Reviewer#1, Concern # 1: It is suggested that the authors provide quantitative descriptions to indicate the advantages of the proposed method in the abstract.

Author response: Thank you for the comment. Unfortunately, the maximum size of the abstract is limited by journal templates, so it is not possible to expand it too much. Nevertheless, the dimensions of objects that can be monitoring with the developed method have been clarified.

Author action: Some details has been added into the abstract.

Reviewer#1, Concern # 2: The introduction is tediously long and confusing. It is recommended to simplify this section.

Author response: Thank you for the comment. According to the authors, a longer introduction allows you to become familiar with the specificity of the measurement method and the nature of the measured objects. Nevertheless, as suggested by the reviewer, the content was shortened.

Author action: The introduction has been shortened.

Reviewer#1, Concern # 3: Provide statistical indicators of displacement estimation results, such as error standard deviation.

Author response: Thank you for the comment. Estimated measurement uncertainty values are presented in paper. The measurement uncertainty has been determined according the rules presented in “Evaluation of measurement data — Guide to the expression of uncertainty in measurement” (Document produced by Working Group 1 of the Joint Committee for Guides in Metrology (JCGM/WG 1). Application of these general principles to the presented measurement method have been presented in another article: Skibicki, J. D. The Issue of Uncertainty of Visual Measurement Techniques for Long Distance Measurements Based on the Example of Applying Electric Traction Elements in Diagnostics and Monitoring. Measurement 2018, 113, 10–21. https://doi.org/10.1016/j.measurement.2017.08.033. Both publications have been cited, as [40] and [41].

Author action: -

Reviewer#1, Concern # 4: From Fig. 16, the fitting relationship curve moves to the left as the temperature increases. There is a cross between the red line and the black line. Explain the possible reasons.

Author response: Thank you for the comment. Changes in natural frequency as a function of contact wire temperature changes are very subtle, especially for higher temperatures. Measurement uncertainties are the most likely cause of the indicated problem. However, the exact cause is unknown and additional research is needed to determine it.

Author action: The content of the paper has been supplemented with an additional comment.

Reviewer#1, Concern # 5: What means did the authors get the natural frequency? FFT or Welch?

Author response: Thank you for the comment. In order to obtain the natural frequency of vibrations, the step response of the system was recorded. The Fourier analysis of displacement waveforms were performed by using the FFT algorithm.

Author action: -

Reviewer#1, Concern # 6: The conclusion is also suggested to be simplified.

Author response: Thank you for the comment. The conclusion has been revised and, as suggested shortened.

Author action: The conclusion has been shortened.

Reviewer 2 Report

The article presents a new vision monitoring method of overhead contact line displacement, which utilizes a set of LED light points, installed along it.

Given my limited background on the topic, I maybe would have liked more than others to understand the shortcomings overcome with the new approach they propose.

For such a reason I suggest enlarging the discussion with respect to this point.

However, the paper is interesting, sound, and well-written and definitely falls within the scope of the journal.

Author Response

Reviewer#2, Concern # 1: The article presents a new vision monitoring method of overhead contact line displacement, which utilizes a set of LED light points, installed along it.

Given my limited background on the topic, I maybe would have liked more than others to understand the shortcomings overcome with the new approach they propose.

For such a reason I suggest enlarging the discussion with respect to this point.

However, the paper is interesting, sound, and well-written and definitely falls within the scope of the journal.

Author response: Thank you for the valuable comment. The main advantage of the proposed measurement method is the possibility of simultaneous (synchronous) observation of displacements of large physical object elements thanks to the use of multi-point observation. The paper presents the method based on the observation of the contact wire. In practice, it can also be used to observe other complex physical objects. Using additional recording devices (e.g. DAQ) you can synchronously record other quantities, such as force, current, voltage, etc. The content of the entire paper has been reviewed and improved so that its comprehensibility has certainly increased.

Author action: The content of the paper has been corrected.

Reviewer 3 Report

This paper proposes the innovative method to perform space-time monitoring of lateral displacement of a contact wire with the use of miniature light points and a vision camera, along with relevant novel software. 

However, som additional clarifications are necessary, such as:

- explaining all variables in eq. 2 and 3.

- give a brief overview of the methods you use as well as the software, because for example the Runge-Kutta numerical method implemented in the Matlab environment is only mentioned in line 424.

- if your application - in which programming language it was developed for which equipment

Sincerely
